# Cardiac neural crest contributes to cardiomyocytes in amniotes and heart regeneration in zebrafish

**Weiyi Tang†, Megan L Martik†, Yuwei Li, Marianne E Bronner***

Division of Biology and Biological Engineering, California Institute of Technology, Pasadena, United States

**Abstract** Cardiac neural crest cells contribute to important portions of the cardiovascular system including the aorticopulmonary septum and cardiac ganglion. Using replication incompetent avian retroviruses for precise high-resolution lineage analysis, we uncover a previously undescribed neural crest contribution to cardiomyocytes of the ventricles in *Gallus gallus*, supported by *Wnt1-Cre* lineage analysis in *Mus musculus*. To test the intriguing possibility that neural crest cells contribute to heart repair, we examined *Danio rerio* adult heart regeneration in the neural crest transgenic line, *Tg(−4.9sox10:eGFP)*. Whereas the adult heart has few *sox10+* cells in the apex, *sox10* and other neural crest regulatory network genes are upregulated in the regenerating myocardium after resection. The results suggest that neural crest cells contribute to many cardiovascular structures including cardiomyocytes across vertebrates and to the regenerating heart of teleost fish. Thus, understanding molecular mechanisms that control the normal development of the neural crest into cardiomyocytes and reactivation of the neural crest program upon regeneration may open potential therapeutic approaches to repair heart damage in amniotes.
DOI: https://doi.org/10.7554/eLife.47929.001

**\*For correspondence:**
mbronner@caltech.edu

†These authors contributed equally to this work

## Introduction

The neural crest is an important stem cell population characterized by its multipotency, migratory behavior, and broad ability to differentiate into derivatives as diverse as elements of the cardiovascular system, craniofacial skeleton, and peripheral nervous system. However, not all neural crest cells are alike, with distinct populations existing along the body axis. One of the most unique neural crest populations is the 'cardiac neural crest' that contributes to the outflow septum and smooth muscle of the outflow tract of the heart. Ablation studies in chick embryos show that removal of the cardiac crest results in a broad range of defects, including persistent truncus arteriosus, abnormal myocardium function, and misalignment of the arch arteries (*Kirby et al., 1983*; *Waldo et al., 1999*; *Bockman et al., 1987*). These defects are highly reminiscent of some of the most common human congenital heart defects. Importantly, other neural crest populations cannot rescue the effects of cardiac neural crest ablation even when grafted in its place, exemplifying the uniqueness of this population (*Kirby, 1989*).

Classically, quail-chick transplantation experiments have been used to uncover contributions of the cardiac neural crest to the heart, with some more recent attempts using antibody staining of migratory neural crest cells or LacZ retroviral lineage analysis as well as transgenic lines such as *Wnt1-Cre* driven β-galactosidase in mammals (*Kirby et al., 1983*; *Kuratani and Kirby, 1991*; *Boot et al., 2003*; *Jiang et al., 2000*). The results suggest that the cardiac neural crest contributes to smooth muscle cells lining the great arteries, outflow tract septum and valves, mesenchyme that remodels pharyngeal arch arteries, and parasympathetic innervation of the heart, such as the cardiac

**eLife digest** Before birth, unspecialized stem cells go through a process called differentiation to form the many types of cells found in the adult. Neural crest cells are a group of these stem cells found in all animals with backbones (i.e. vertebrates) including humans. These cells migrate extensively during development to form many different parts of the body. Due to their contributions to diverse organs and tissues, neural crest cells are very important for healthy development.

The heart ventricle is one of the tissues to which neural crest cells contribute during embryonic development in fish and amphibians. However, it was unclear whether this is also the case for birds or mammals or whether neural crest cells have any roles in the regeneration of the adult heart after injury in fish and amphibians.

To address these questions, Tang, Martik et al. used cell biology techniques to track neural crest cells in living animals. The experiments revealed that neural crest cells contribute to heart tissue in developing birds and mammals and help repair the heart in adult zebrafish. Further results showed that the contribution of neural crest cells to the heart is controlled by the same genes during both the growth of the embryonic heart and the repair of the adult heart.

These results provide new insights into the repair and healing of damaged heart muscle in fish. They also show that similar processes could exist in mammals, including humans, suggesting that activating neural crest cells in the heart could treat damage caused by heart attacks and related conditions.

DOI: https://doi.org/10.7554/eLife.47929.002

ganglion. However, inconsistencies remain between different lineage approaches, most of which suffer from high background and low cellular resolution.

To reconcile these differences, here, we use a multi-organismal approach to examine the lineage contributions of cardiac neural crest to the heart. Using a novel retroviral labeling approach in chick and confirmed by *Wnt1-Cre* reporter lines in mouse, we reveal a previously undetected contribution of the amniote cardiac neural crest to the trabecular myocardium of the ventricles, a derivative previously thought to be confined to non-amniotic vertebrates (*Sato and Yost, 2003*; *Li et al., 2003*; *Cavanaugh et al., 2015*).

The homologous cardiac neural crest contribution to cardiomyocytes across diverse species raised the intriguing possibility that these cells may contribute to cardiac repair. As the adult zebrafish heart exhibits extensive regenerative capacity, we turned to this model to test whether the neural crest may contribute to heart regeneration (*Poss et al., 2002*). Intriguingly, we show that resected adult zebrafish hearts reactivate many genes of a neural crest gene regulatory program during the regeneration process. Taken together, these results demonstrate an evolutionarily conserved contribution of neural crest cells to cardiomyocytes across vertebrates and a previously unappreciated role during heart regeneration.

## Results

### Labeling the chick cardiac neural crest using Replication Incompetent Avian retrovirus

To specifically label cardiac neural crest cells prior to their emigration from the neural tube and identify novel progeny of chick cardiac crest, we use a replication-incompetent avian retrovirus (RIA) that indelibly and precisely marks neural crest progenitors for long term lineage analysis at single cell resolution and without the need for tissue grafting. To this end, the post-otic neural tube of the hindbrain adjacent to somites 1–3 was injected at Hamburger and Hamilton (HH) stage 9–10 with high-titer ($1 \times 10^7$ ifu/mL) RIA (*Figure 1A*), which drives expression of nuclear localized *H2B-YFP* under control of a constitutive RSV promoter (*Li et al., 2017*; *Tang et al., 2019*; *Fields-Berry et al., 1992*; *Chen et al., 1999*; *Hamburger and Hamilton, 1951*). At this stage in the development, premigratory cardiac neural crest cells are positioned within the dorsal neural tube and about to emigrate. Accordingly, this labeling approach solely marks hindbrain neural tube cells including premigratory

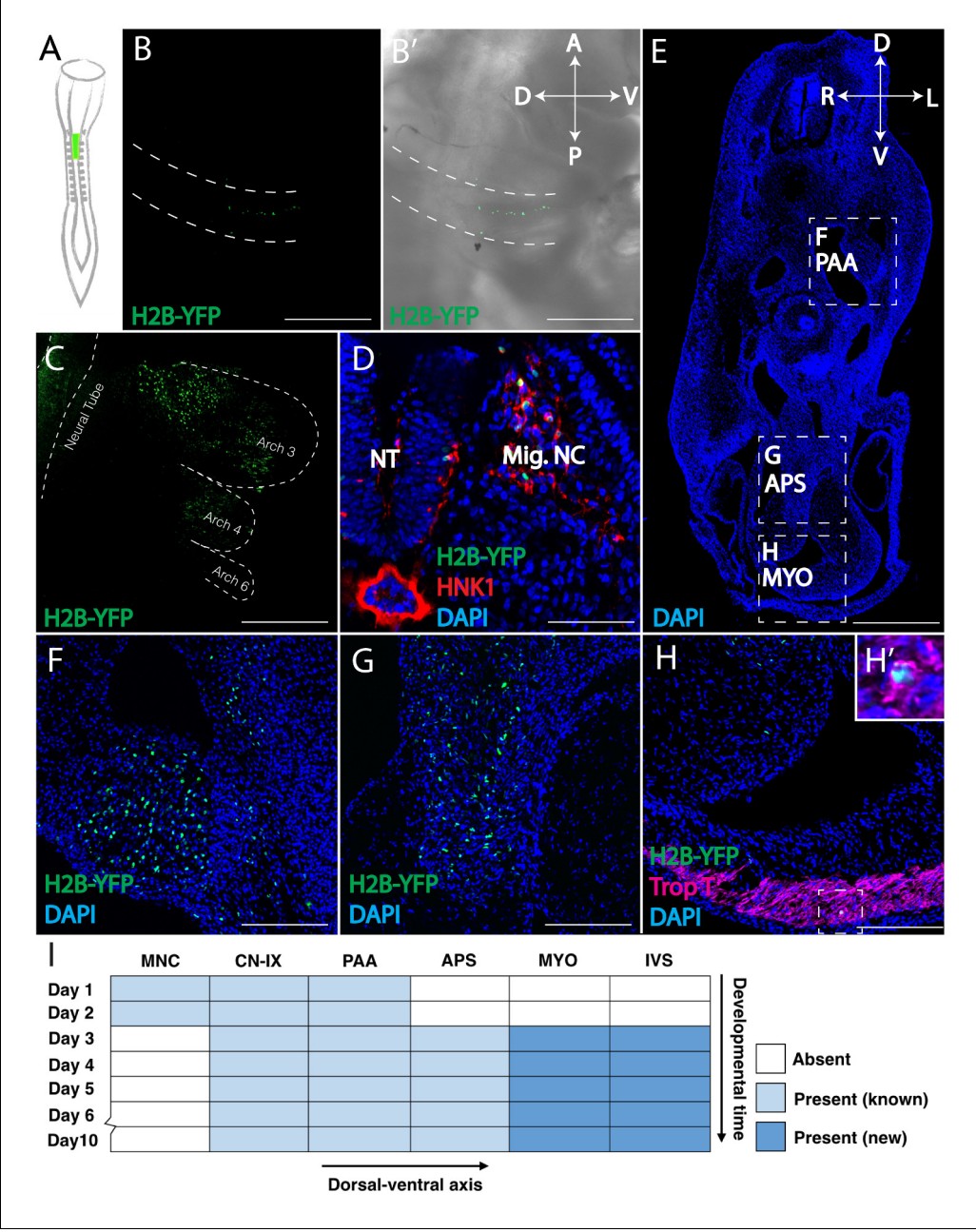

**Figure 1.** Retrovirally mediated fate mapping of cardiac neural crest reveals novel derivatives. (**A**) Schematic diagram of the approach: Replication Incompetent Avian (RIA) retrovirus encoding nuclear H2B-YFP was injected into the lumen of the hindbrain from which cardiac neural crest arises. (**B**) One day post-infection (HH14), whole mount image (lateral view) showing virally labeled progeny (green) in the cardiac migration stream en route to pharyngeal arch 3. (**B'**) Brightfield image to show anatomical information. A, anterior; P, posterior; D, dorsal; V, ventral. (**C**) Two days post-infection (HH18), virally labeled cardiac crest has populated pharyngeal arches 3, 4 and 6, highlighted with dashed line. (**D**) Transverse section showing that labeled cardiac crest expresses neural crest marker HNK-1 (red). D, dorsal; V, ventral; L, left; R, right. (**E**) Low magnification transverse section of an E6 embryo (DAPI, blue). Dashed boxes show relative positions of cardiac crest-derived populations. (**F–H**) High magnification image of selected regions in E: pharyngeal arch arteries (**F**); aorticopulmonary septum (**G**); Neural crest derivatives located in the outflow tract express Troponin T (magenta), a myocardium marker (**H, H'**). (**I**) Temporal map of the establishment of distinct cardiac neural crest derivatives. Labeled cells initially are in the migration stream, cranial nerve IX (CN-IX) and mesenchyme around pharyngeal arch arteries (PAA). Subsequently, they populate the aorticopulmonary septum (APS), myocardium (MYO) and interventricular septum (IVS). Separate channels are

*Figure 1 continued on next page*

*Figure 1 continued*

displayed in *Figure 1—figure supplement 1*. Light blue indicates known neural crest derivatives. Dark blue reflects newly discovered neural crest derivatives. Scale bars: B, C, E 400 µm; D, F, G, H 100 µm.
DOI: https://doi.org/10.7554/eLife.47929.003
The following figure supplement is available for figure 1:

**Figure supplement 1.** Separate channels for retroviral lineage analysis and immunohistochemistry.
DOI: https://doi.org/10.7554/eLife.47929.004

cardiac neural crest cells that subsequently delaminate from the dorsal neural tube during a two-hour time window when the virus remains active.

Virally infected embryos were then allowed to develop for 1–9 days post injection, cryo-sectioned, and analyzed using confocal microscopy. One day after injection, whole mount imaging revealed RIA-labeled cells migrating in a stream along pharyngeal arch 3 (*Figure 1B,B'*), that subsequently accumulated in pharyngeal arches 3, 4 and 6 two days after infection (*Figure 1C*). Next, we confirmed that all labeled cells in the periphery co-localized with the migratory neural crest marker, HNK-1, demonstrating that the neural crest is the only population labeled with H2B-YFP outside the neural tube, thus verifying specificity of infection (*Figure 1D*, *Figure 1—figure supplement 1A*). With time, labeled cardiac crest cells were observed in numerous and diverse derivatives, populating the cardiovascular system in a proximal to distal progression (*Figure 1E–I*, *Supplementary file 1a*). Consistent with quail-chick chimera, we observed RIA-labeled cells adjacent to and within the walls of pharyngeal arch arteries, in the aorticopulmonary septum, outflow tract, and cardiac cushion. Moreover, we definitively observed YFP-labeled cells in the superior interventricular septum, a site for which the neural crest contribution has been controversial, although ventricular septal defects are common after cardiac neural crest ablation (*Kirby et al., 1985*). The cells of the outflow tract septum and pharyngeal arch arteries differentiated into smooth muscle actin (SMA) positive cells on embryonic day (E) 5 (*Figure 2A,B*).

Importantly, by E3 and onward, virally labeled neural crest cells were observed in the myocardium of both the outflow tract and the ventricles, where they expressed the myocardial markers, Troponin T and Myosin Heavy Chain (*Figure 1H*, *Figure 1—figure supplement 1B*, outflow tract; *Figure 2C, D*, *Figure 2—figure supplement 1A,B*, ventricles). These neural crest-derived cardiomyocytes were not actively undergoing cell division or programmed cell death (*Figure 2E,F*), consistent with the stable presence of cells observed over time (*Figure 1I*, *Supplementary file 1a,1b*). *Supplementary file 1a and 1b* present quantification of contributions of virally labeled cells in the chick ventricular myocardium. While previous lineage tracing experiments in zebrafish showed that a stream of neural crest cells integrate into the myocardium of the primitive heart tube to give rise to cardiomyocytes, our results present the first evidence of a homologous neural crest contribution to cardiomyocytes in chick embryos (*Sato and Yost, 2003*; *Li et al., 2003*; *Cavanaugh et al., 2015*).

## Lineage analysis in the mouse embryo

To test whether the contribution of cardiac neural crest cells to the myocardium was conserved in mammals, we examined *Wnt1-Cre;ZsGreen^{fl/fl}* transgenic mice in which neural crest cells were labeled with cytoplasmic GFP (*Chai et al., 2000*). Embryos were fixed at E15.5 (similar to E7 in chick). Analogous to the results in chick embryos, we observed a large number of ZsGreen-positive myocardial cells in the outflow tract and ventricles, as confirmed by Troponin T expression (*Figure 3A–C*). To avoid ectopic expression that has been associated with the *Wnt1-Cre;ZsGreen^{fl/fl}* transgenic line due to endogenous *Wnt1* activation caused by in-frame ATG located upstream of *Wnt1* start codon, we tested an improved Wnt1 line (*Wnt1-Cre2+; R26mTmG* mouse line) without ectopic activation of canonical *Wnt/β-catenin* pathway (*Lewis et al., 2013*). The results were similar to those observed with the *Wnt1-Cre;ZsGreen^{fl/fl}* transgenic mice (*Figure 3D,E*). As in the chick embryos, murine neural crest derived cells were present in the outflow tract, interventricular septum, and myocardium of both ventricles.

The numbers of neural crest-derived cells appear to decrease with distance along the proximal-to-distal axis (*Figure 3—figure supplement 1A*), such that no neural crest-derived cardiomyocytes were observed in the apex of the heart (*Figure 3—figure supplement 1D,E*). As in the chick, the

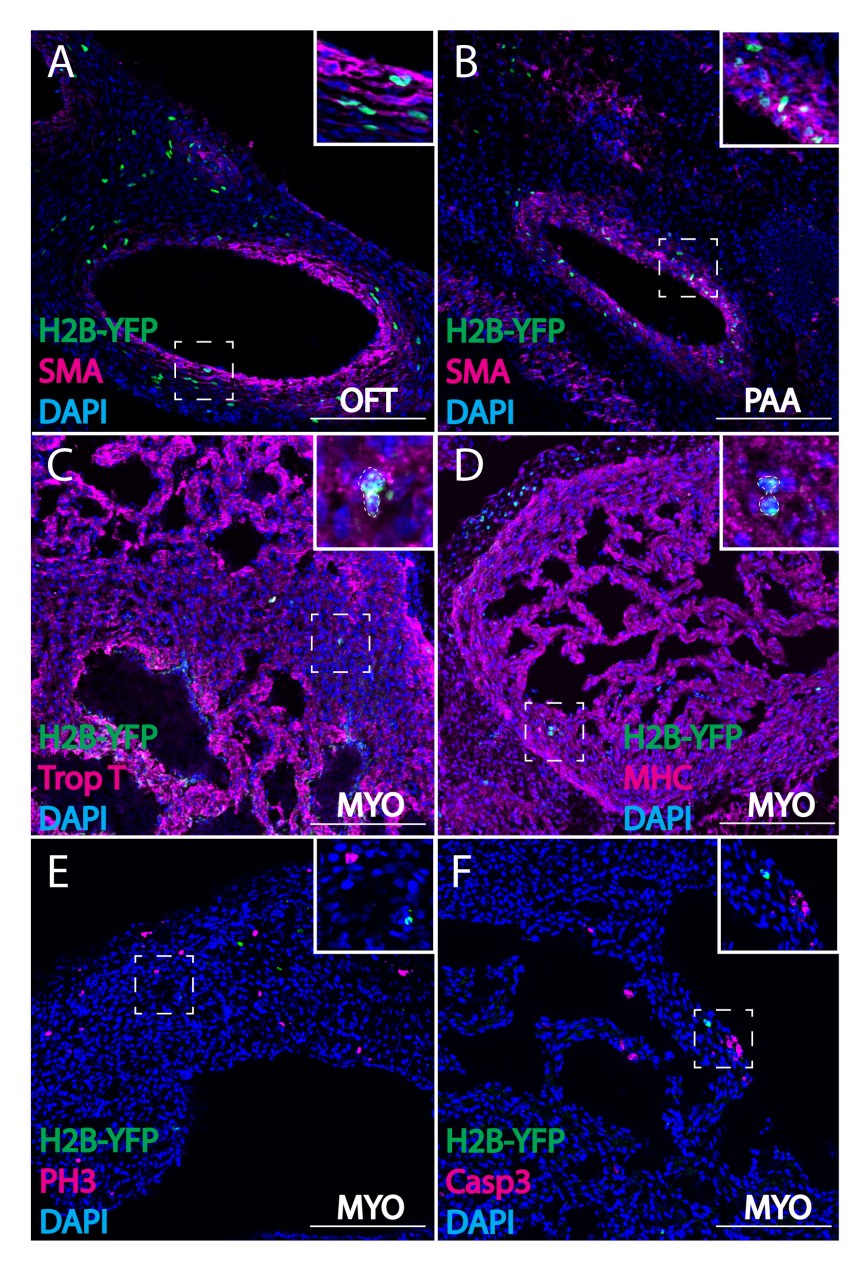

**Figure 2.** Cardiac crest-derived cells differentiate into smooth muscle and cardiomyocytes in avian embryos. (A, B) Retrovirally labeled cardiac crest cells (H2B-YFP, green) that migrate into the outflow tract (A, OFT) and pharyngeal arch arteries (B) express smooth muscle actin (SMA, magenta) marker. (C, D) Labeled cardiac crest cells that enter the ventricle express myocardial marker Troponin T (magenta) (C), and myocardial terminal differentiation marker Myosin Heavy Chain (MHC, magenta) (D) enclosed in dashed line. (E, F) Neural crest-derived cardiomyocytes are not actively dividing or undergoing apoptosis, as demonstrated by phosphohistone H3 staining (PH3, magenta) (E) and Caspase 3 staining (magenta) (F). Transverse view of E6 embryos. Separate channels are displayed in supplement 1. Scale bars: 100 μm.

DOI: https://doi.org/10.7554/eLife.47929.005

The following figure supplement is available for figure 2:

**Figure supplement 1.** Separate channels for retroviral lineage analysis and immunohistochemistry.
DOI: https://doi.org/10.7554/eLife.47929.006

numbers of *Wnt1+* cells remain stable with time, and the cells do not appear to undergo active cell division or apoptosis (*Figure 3—figure supplement 1B,B',C,C'*). This contribution persists postnatally, as *Wnt1+* cells are present at postnatal day 2 (*Figure 3—figure supplement 1F–H*). These results are consistent with previous studies using less specific P0-cre lines and demonstrate that comparable cardiac crest contributions occur in birds and mammals (*Tomita et al., 2005*; *Tamura et al., 2011*). Quantification of numbers of neural crest lineage labeled cells in the trabeculated myocardium of mice reveals that they represent approximately 17% of the population in the proximal half of the ventricle (*Supplementary file 1a*).

The lineage contributions of neural crest-derived cells in chick and mouse are remarkably similar to those previously shown in zebrafish (*Sato and Yost, 2003*; *Li et al., 2003*; *Cavanaugh et al., 2015*). In all three species, neural crest cells contribute to cardiomyocytes of the trabecular myocardium. This homologous lineage contribution in both amniotes and anamniotes raised the intriguing possibility that neural crest cells may represent a cell population that could contribute to heart repair in adults.

## Reactivation of neural crest gene regulatory genes during adult zebrafish heart regeneration

In adult birds and mammals, cardiac injury leads to scarring with little regeneration, whereas heart regeneration is common in amphibians and fish (*González-Rosa et al., 2017*). For example, adult zebrafish have the capacity to regenerate their hearts after removal of up to 20% of the ventricle. This has been shown to occur by dedifferentiation and proliferation of pre-existing cardiomyocytes (*Poss et al., 2002*; *Jopling et al., 2010*). Given that cardiac neural crest cells give rise to a portion of zebrafish cardiomyocytes during development (*Sato and Yost, 2003*; *Li et al., 2003*; *Cavanaugh et al., 2015*) similar to those we report here in chick and mouse, we next asked whether

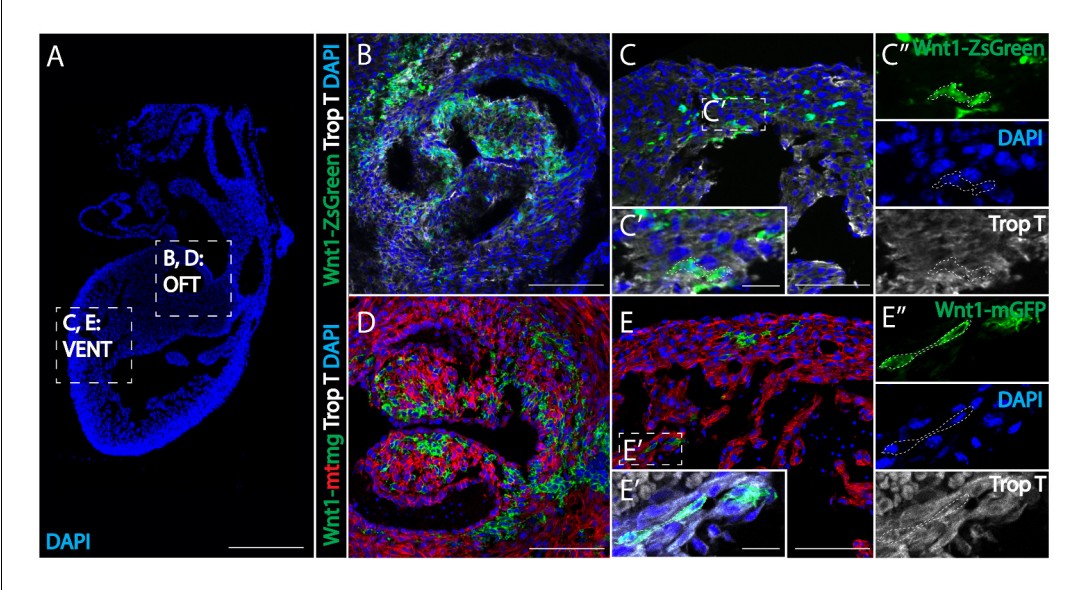

**Figure 3.** *Wnt1-Cre* fate mapping in mice confirms the presence of cardiac crest-derived myocardium. (**A**) Low magnification image to show the relative anatomical positions of a mouse heart at E15.5 (sagittal view, DAPI-blue). (**B, C**) In *Wnt1-Cre; ZsGreen*<sup>fl/fl</sup> mice, neural crest-derived cells (green, *Wnt1-Cre* driven ZsGreen expression is abbreviated as Wnt1-ZsGreen, enclosed in dashed line) were observed in myocardium (Troponin T, gray) of the outflow tract (**B**), and ventricle (VENT) (**C**, **C''**: separate channels of inset C'). (**D, E**) Similar results were obtained from *Wnt1-Cre2+; R26mTmG* mice (*Wnt1-Cre2+* driven replacement of membrane localized tdTomato (mT) by EGFP (mG) (abbreviated as Wnt1-mtmg), where cardiac crest-derived cells (green, enclosed in dashed line) were present in myocardium of the outflow tract (**D**) and ventricle (Troponin T, gray) (**E**, **E''**: separate channels of inset E'). Transverse view. Spatial-temporal information and antibody staining are displayed in supplement 1. Scale bars: A 400 µm; B-E 100 µm; C', E' 10 µm.
DOI: https://doi.org/10.7554/eLife.47929.007

The following figure supplement is available for figure 3:

**Figure supplement 1.** Spatial-temporal distribution of neural crest-derived cardiomyocytes in *Wnt1-Cre* mouse.
DOI: https://doi.org/10.7554/eLife.47929.008

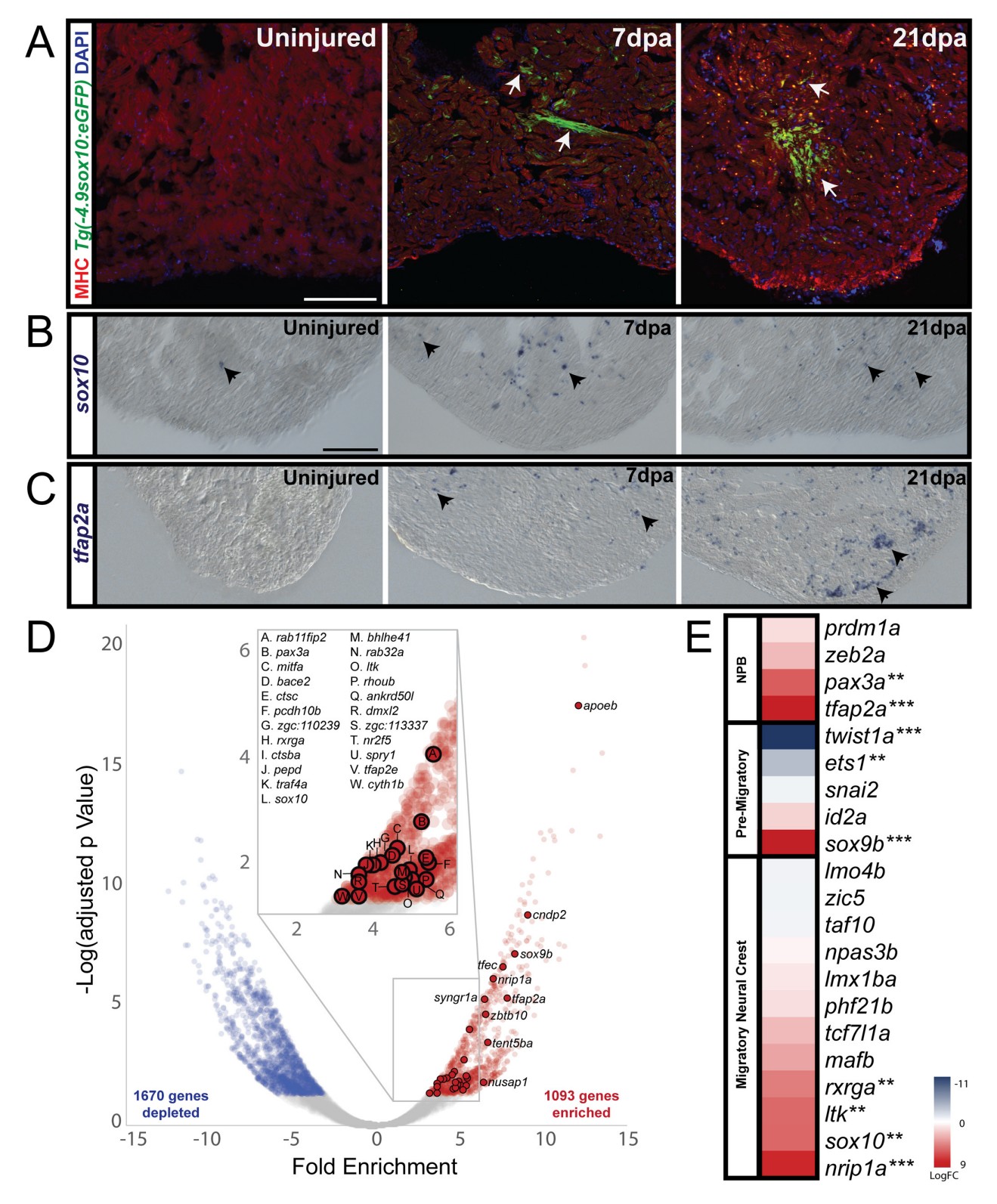

**Figure 4.** Cardiac neural crest contributes to heart regeneration in zebrafish. (**A**) In sham-operated adult zebrafish hearts from a transgenic line expressing GFP under the control of a sox10 promoter, very few cells expressed *Tg(−4.9sox10:eGFP)* (green) (n = 3). 7 days-post amputation (dpa), the *sox10* promoter was reactivated as shown by GFP+ cells in the trabeculated myocardium near the site of injury (*Tg(−4.9sox10:eGFP)*, green) (n = 6). 21dpa, when the resected apex regenerated, more GFP+ cells were observed in sagittal sections within and surrounding the site of injury (n = 6).

*Figure 4 continued on next page*

*Figure 4 continued*

Sections in A are counterstained with DAPI in blue and Myosin Heavy Chain in red. (**B**) Endogenous *sox10* mRNA expression was observed by paraffin section in situ hybridization in uninjured, 7dpa, and 21dpa hearts. Arrows denote cells with *sox10* expression. From these results, we conclude that *sox10* is reactivated after injury. (**C**) Along with *sox10*, expression of neural crest marker, *tfap2a*, was also enriched after injury. Arrows label areas of expression in the myocardium. (**D**) Differential gene expression analysis of FACS-sorted *Tg(sox10:mRFP)+* and FACS-sorted *Tg(sox10:mRFP)-* transcriptomes show n = 1093 genes are enriched at 21dpa in the *sox10+* cells compared to the rest of the ventricular tissue (n = 12 ventricles per replicate). Zebrafish neural crest genes as determined by GO analysis are highlighted on the volcano plot. (**E**) Upregulation of neural crest gene regulatory network genes was also observed from our differential expression analysis (**p<0.05, ***p<0.001). Co-localization of *sox10* mRNA expression with *Tg(sox10:GAL4-UAS-Cre;ubi:Switch)+* neural crest-derived cardiomyocytes is presented in supplement 1; schematic diagram of experimental design for obtaining the regenerating neural crest transcriptome and further analysis of gene enrichments is presented in supplement 2. Scale bars: 100 µm.

DOI: https://doi.org/10.7554/eLife.47929.009

The following figure supplements are available for figure 4:

**Figure supplement 1.** Co-localization of sox10 mRNA expression with *Tg(sox10:GAL4-UAS-Cre;ubi:Switch)+* neural crest-derived cardiomyocytes.

DOI: https://doi.org/10.7554/eLife.47929.010

**Figure supplement 2.** Analysis of regenerating neural crest transcriptome.

DOI: https://doi.org/10.7554/eLife.47929.011

the progeny of these cells might have the ability to contribute to heart regeneration in adult zebrafish.

To address this possibility, we first turned to a transgenic line expressing GFP under the control of a *sox10* promoter, *Tg(−4.9sox10:eGFP)*, that labels all embryonic migratory neural crest lineages to address whether neural crest-derived cardiomyocytes reactivated their developmental program upon injury (*Carney et al., 2006*).

While *sox10* is expressed in migrating zebrafish cardiac neural crest cells, it is down-regulated in the embryo shortly after these cells reach the heart (*Cavanaugh et al., 2015*). We confirmed this in adult hearts, finding that very few cells within the apex of the adult myocardium of control adult fish expressed *sox10* one month post-sham injury, in which the body cavity was opened but no resection was made (*Figure 4A*, *Supplementary file 1c*, n = 3). However, after surgical removal of ~20% of the ventricular apex, cells in the heart reactivated the *sox10* promotor sequence and began to re-express GFP in cardiomyocytes of the trabeculated myocardium near the injured site by 7 days post resection (dpa) (*Figure 4A*; n = 6). GFP expression was not limited to the regenerating tissue but was also observed in the uninjured part of the ventricle. By 21dpa, the hearts had undergone vast regeneration and morphologically were nearly indistinguishable from controls (*Figure 4A*; n = 6). Interestingly, consistent with our prediction, the regenerating apex was comprised of more *sox10+* positive cells (*Figure 4A,B*, *Supplementary file 1c*), suggesting that these cells had proliferated and redeployed a neural crest gene regulatory program during the heart regeneration process.

To test if *sox10* and other bona fide neural crest markers such as *tfap2a*, were upregulated endogenously, we performed in situ hybridization on paraffin sections of regenerating and uninjured ventricles. The results reveal upregulation of expression of *sox10* and *tfap2a* transcripts after injury, whereas they were mostly absent from uninjured ventricles (*Figure 4B,C*). Furthermore, we observed co-localization of *sox10* transcripts with a *Tg(sox10:GAL4-UAS-Cre;ubi-Switch),* which permanently labels all *sox10*-derived lineages with mCherry (*Figure 4—figure supplement 1*, n = 2). The *Tg(sox10:GAL4-UAS-Cre;ubi-Switch)* is a double transgenic line for the *sox10:GAL4-UAS-Cre* transgene and the *ubi:Switch* reporter in which the *sox10* promoter drives expression of Cre recombinase. Upon activation of *sox10* expression in neural crest cells, eGFP is excised and so cells of the *sox10* lineage are permanently labeled with mCherry (*Cavanaugh et al., 2015*). All cells expressing *sox10+* transcripts also had mCherry, though not all mCherry positive cells were *sox10+* at the 7 day time point (*Figure 4—figure supplement 1*, insets 1 and 2). Our results are consistent with recent findings from Abdul-Wajid and colleagues, who observed that ablation of the embryonic neural crest yields few or no *sox10+* cells in the adult heart and results in severe heart defects (*Abdul-Wajid et al., 2018*). This suggests there are no subsequent post-embryonic neural crest additions to the heart and that the population we observe re-expressing neural crest genes are embryonic-derived neural crest progeny.

These results raise the intriguing possibility that the neural crest developmental gene regulatory network was being redeployed in neural crest-derived cells of the heart during regeneration. To test this, we performed transcriptional profiling of *sox10:mRFP+* cells in the regenerating zebrafish hearts at 21dpa. To this end, we dissected and dissociated injured ventricles (n = 12 per replicate) into single cell suspensions and performed FAC-sorting of *sox10:mRFP+* cells (*Figure 4—figure supplement 2A*). The results were compared with mRFP negative cardiac cells from the same injured, isolated ventricles. This led to the identification of 1093 genes that are significantly enriched (p-adj <0.05) in regenerating *sox10+* cells compared to *sox10-* cells of the same injured ventricles (*Figure 4*, *Figure 4—figure supplement 2*). We then compared the differentially expressed genes of isolated 21dpa *sox10+* cells to: 1) our recently published chick developmental cardiac neural crest gene regulatory program, 2) known zebrafish neural crest genes, and 3) core neural crest gene regulatory network genes expressed at all axial levels (*Tani-Matsuhana et al., 2018*; *Martik and Bronner, 2017*; *Lukoseviciute et al., 2018*). The results revealed upregulation of many genes of the embryonic neural crest gene regulatory network at the time of regeneration (*Figure 4D and E*).

Interestingly, numerous genes known to be responsible for cardiomyocyte proliferation also are expressed in *sox10+* cells upon heart injury (*Figure 4—figure supplement 2E*) (*González-Rosa et al., 2017*). The co-expression of these genes as well as an upregulation of a cell proliferation gene signature suggests a role for sox10-derived cells in cardiomyocyte proliferation during regeneration (*Figure 4—figure supplement 2C*). Furthermore, these results suggest that the population of proliferating cardiomyocytes in the regenerating heart is heterogeneous and comprised of both neural crest- and mesoderm-derived cardiomyocytes (*González-Rosa et al., 2017*; *Sánchez-Iranzo et al., 2018*; *Schindler et al., 2014*; *Kikuchi et al., 2010*).

## Discussion

While much attention has been paid to the molecular signals that promote myocardial dedifferentiation and proliferation during regeneration, far less is known about the cell lineages that contribute to the regeneration process. Based on our observation on the lineage relationship between cardiac neural crest cells and cardiomyocytes during development, we propose that neural crest-derived cells (progenitors and/or pre-existing cardiomyocytes) may represent a key population that proliferates and differentiates into new cardiomyocytes after injury.

Our cell lineage labeling results provide direct evidence for a neural crest contribution to the undamaged myocardium of the amniote heart. Furthermore, consistent with previous lineage tracing experiments in zebrafish (*Cavanaugh et al., 2015*), where a proportion of cardiac crest derived-cells were located in the trabeculated myocardium in adult fish, we show that after injury, there is activation of numerous neural crest gene regulatory transcription factors and other neural crest genes during regeneration (*Figure 4*). While the underlying gene regulatory network of neural crest cells is responsible for formation of cardiomyocytes during normal development, we speculate that it also does so in a similar manner upon injury by redeploying *sox10* and other neural crest gene regulatory network genes. The finding that *sox10*-derived cells are primarily in the proximal trabecular myocardium of the zebrafish heart suggests that these cells must be migrating into to the wound site after injury. Of course, we cannot rule out the possibility that the cells that reactivate *sox10* and the neural crest program may come from another adult lineage. But in the adult, their molecular signature strongly correlates with that of embryonic neural crest cells (*Figure 4D and E*). Whereas our data clearly show that the *sox10+* cells contribute to cardiomyocytes (*Figure 4A*), whether they also might contribute to other lineages (e.g. hematopoietic cells) within the regenerated tissue remains to be explored.

Why was the contribution of neural crest cells to cardiomyocytes in amniotes previously missed? Interspecific quail-chick chimera are generated via transplantation of donor tissue into the host, which requires time to heal (*Kirby et al., 1983*). If the neural crest cells that migrate to the ventricles are the earliest migrating cells, this population may have been delayed after grafting due to wound healing and hence unable to migrate as far. Alternatively, the labeled cells may have been missed since it can be challenging to identify a small population of dispersed quail cells amongst many more numerous chick cells. Furthermore, cell behavior might be altered when transplanted quail cells are introduced into a chick environment. Our retroviral lineage labeling circumvents these issues by indelibly labeling an endogenous neural crest population without the need for grafting.

Moreover, the labeled cells are easily detectable due to their fluorescent readout. For lineage labeling in mice, there were hints in the literature regarding a possible neural crest contribution to cardiomyocytes. However, the experiments were either indirect or used lineage tracing techniques that were not specific to the neural crest. For example, Tomita et al. showed that cells isolated from 'cardiospheres' can behave like neural crest cells when injected into chick embryos (*Tomita et al., 2005*). In addition, lineage analysis in mouse using a P0-cre line revealed EGFP-positive cells in the myocardium that gather at the ischemic border upon injury (*Tomita et al., 2005*). However, P0 is not a neural crest specific marker, making these results inconclusive at the time. In contrast, Wnt1 is the 'gold standard' for neural crest labeling and the improved Wnt1 line (*Wnt1-Cre2+; R26mTmG*) corrects possible ectopic expression problematic in the original *Wnt1-Cre;ZsGreen line* (*Jiang et al., 2000*; *Chai et al., 2000*; *Lewis et al., 2013*).

In chick and mouse, neural crest-derived cells comprise a significant portion (~17%) of the trabeculated myocardium in the proximal part of both ventricles. Interestingly, this percentage is similar to what has been reported in zebrafish (*Cavanaugh et al., 2015*; *Abdul-Wajid et al., 2018*). In amniotes, we find that the density of the cells decreases along the proximal-distal axis and appears to be stable through time (*Figure 1I*, *Supplementary file 1a,b*). The presence of neural crest-derived cardiomyocytes across vertebrates and the redeployment of a *sox10+* cell population in zebrafish heart regeneration suggest that the neural crest-derived myocardium might also play a role in heart regeneration in neonatal mice, which requires further testing.

In summary, the present results show, for the first time, the common ability of cardiac neural crest cells across diverse vertebrates to contribute to heart muscle. Moreover, these cells appear to be critical for cardiac regeneration in zebrafish. If the results extrapolate to other species, the mechanisms that control the normal development of the neural crest into cardiomyocytes may be harnessed to stimulate these cells to proliferate and regenerate new cardiomyocytes, thus offering potential therapeutic approaches to repair heart damage in mammals including humans.

## Materials and methods

### Key resources table

| Reagent type (species) or resource | Designation | Source or reference | Identifiers | Additional information |
|---|---|---|---|---|
| Genetic reagent (*Mus musculus*) | *Wnt1-Cre; ZsGreen*<sup>fl/fl</sup> | PMID:10725243 | Jackson Laboratories, Stock# 003829 | Drs. Xia Han and Yang Chai at University of Southern California, Center for Craniofacial Molecular Biology |
| Genetic reagent (*Mus musculus*) | *Wnt1-Cre2+((129S4-Tg(Wnt1-cre)1Sor/J));R26mTmG* | PMID: 23648512 | Jackson Laboratory, Stock# 22137 | Dr. Jeffrey Bush at University of California, San Francisco |
| Genetic reagent (*Danio rerio*) | *Tg(−4.9sox10:eGFP)* | PMID: 17065232 | ZFIN ID: ZDB-TG CONSTRCT-070117–69 | |
| Genetic reagent (*Danio rerio*) | Tg(sox10:GAL4-UAS-Cre;ubi-Switch) | PMID: 26086691 | | Drs. Ann M. Cavanaugh and Jau-Nian Chen at Department of Molecular, Cell and Developmental biology, University of California, Los Angeles |
| Genetic reagent (*Danio rerio*) | *Tg(sox10:mRFP)* | PMID: 18176560 | ZFIN ID: ZDB-TGCONSTRCT-080321–2 | |
| Cell line (*Galllus gallus* DF1) | UMNSAH/DF-1 fibroblast spontaneously transformed | ATCC | #CRL-12203, Lot number 62712171; RRID:CVCL_0570 | |
| Recombinant DNA reagent | RES-H2B-YFP-DD | Addgene | RRID:Addgene_96893 | |

*Continued on next page*

*Continued*

| Reagent type (species) or resource | Designation | Source or reference | Identifiers | Additional information |
|---|---|---|---|---|
| Antibody | Mouse monoclonal anti-bovine Troponin T, IgG2a (CT3) | DSHB | RRID:AB_528495 | Dilution (1:10) |
| Antibody | Mouse monoclonal anti-chicken Myosin Heavy Chain, IgG1 kappa light chain (ALD58) | DSHB | RRID:AB_528361 | Dilution (1:10) |
| Antibody | Mouse monoclonal anti-chicken Myosin Heavy Chain, IgG1 kappa light chain (F59) | DSHB | RRID:AB_528373 | Dilution (1:10) |
| Antibody | Mouse monoclonal anti-NH2 terminal synthetic decapeptide of alpha smooth muscle actin, IgG2a | Sigma | # A5228 | Dilution (1:500) |
| Antibody | Mouse monoclonal anti-human phospho-histone H3, IgG1 | Abcam | #ab14955 | Dilution (1:500) |
| Antibody | Rabbit polyclonal anti-human Caspase 3, IgG | R and D systems | #AF835 | Dilution (1:500) |
| Antibody | Goat polyclonal anti -GFP, IgG | Abcam | #ab6673 | Dilution (1:500) |
| Antibody | Goat polyclonal anti-mouse IgG2a Alexa-568 | Molecular Probes | RRID:AB_2535773 | Dilution (1:1000) |
| Antibody | Goat polyclonal anti-mouse IgG1 Alexa-568 | Molecular Probes | RRID:AB_2535766 | Dilution (1:1000) |
| Antibody | Goat polyclonal anti-rabbit IgG Alexa-568 | Molecular Probes | RRID:AB_2534121 | Dilution (1:1000) |
| Antibody | Donkey polyclonal anti-goat IgG Alexa-488 | Molecular Probes | RRID:AB_2534102 | Dilution (1:1000) |
| Software, algorithm | Image processing software FIJI | https://imagej.net/Fiji | | |
| Software, algorithm | R v3.6.1 | https://www.r-project.org/ | | |
| Software, algorithm | DESeq2 | PMID: 25516281 | RRID:SCR_015687 | |
| Software, algorithm | Bowtie2 | PMID: 22388286 | RRID:SCR_005476 | |
| Software, algorithm | featureCounts (Subread) | PMID: 24227677 | RRID:SCR_009803 | |
| Other | Accumax | Innovative Cell Technologies, Inc | #AM105 | |
| Commercial assay or kit | SMART-seq Ultra Low Input RNA Kit V4 | Takara Clontech | #634891 | |

## Cell culture and retrovirus preparation

Using a standard transfection protocol, chick DF1 cells (ATCC, Manassas, VA; #CRL-12203, Lot number 62712171, RRID:CVCL_0570, Certificate of Analysis with negative mycoplasma testing at the ATCC website) were transfected with RIA-H2B-YFP plasmid (RRID:Addgene_96893) and ENV-A plasmid in 15 cm culture dishes. Cell culture medium was collected 24 hr post-transfection, and twice per day for four days, then centrifuged at 26,000 rpm for 1.5 hr. The supernatant was dried with

aspiration, and the pellet was dissolved in 20–30 µl of DMEM to a final titer of $1 \times 10^7$ ifu/mL. Viral aliquots were stored in −80°C until the time of injection.

## Chick embryo processing and viral injection

Viral stock was diluted 1:2 with Ringer's solution (0.9% NaCl, 0.042%KCl, 0.016%CaCl$_2$ • 2H$_2$O wt/vol, pH7.0) to generate the working solution, which was mixed with 0.3 µl of 2% food dye (Spectral Colors, Food Blue 002, C.A.S# 3844-45-9) as indicator. The lumen of the neural tube adjacent to the middle of the otic vesicle to the level of somite three was injected with 0.2 µl of working in HH8-10 chicken embryos. Embryos were sealed with surgical tap and incubated at 37°C for 1–9 days, harvested at HH14 (n = 5), HH18 (n = 5), HH21 (n = 4), HH25 (n = 4), HH28 (n = 12), HH32 (n = 4) and E10 (n = 4). At the time of harvesting, chick embryos were dissected, fixed in 4%PFA in PBS for 30 mins at 4°C, then embedded in gelatin and sectioned (*Microm* HM550 cryostat).

## *Wnt1-Cre* mouse and tissue preparation

The *Wnt1-Cre; ZsGreen*$^{fl/fl}$ mice described in *Chai et al. (2000)* (gift from Drs. Xia Han and Yang Chai at University of Southern California, Center for Craniofacial Molecular Biology) were harvested and fixed at E15.5 (n = 8) and P2 (n = 2). The hearts were dissected, fixed in 4%PFA in PBS for 30mins at 4°C. E15.5 *Wnt1-Cre2+; R26mTmG* mice (*Lewis et al., 2013*) (129S4-Tg(Wnt1-cre)1Sor/J, gift from Dr. Jeffrey Bush at University of California, San Francisco, n = 3) were fixed with 4% PFA overnight before dissection. The hearts were embedded in gelatin, and sectioned.

## Quantification of neural crest contributions to the ventricular myocardium and regeneration

To quantify RIA-labeled cells in chick embryos, three consecutive sections of the same axial level were imaged per embryo. The number of YFP-positive cells was averaged to account for variability due to sampling. n = 4–6 embryos were analyzed at each stage as biological replicates. The results are presented as presence or absence of virally labeled cardiac neural crest derivatives at different anatomical locations in *Figure 1I* and as numerical values in *Supplementary file 1a, 1b*. To quantify Wnt1-Zsgreen+ cells in E15.5 mouse heart, three consecutive sections of the same axial level were imaged per embryo (n = 4). Automated particle analysis was conducted with FIJI program to estimate the total number of Zsgreen+ cells in the image. For the percentage of neural crest-derived cells in the ventricle, the same procedure was performed with the DAPI channel which represents total cell population. % Zsgreen/DAPI was calculated, and averaged to the result presented in the text of *Supplementary file 1a*. Same analysis was conducted to estimate the number of *sox10:eGFP* + cells in 7dpa (n = 3), 21dpa (n = 3) and sham operated (n = 3) hearts in an area of $2 \times 10^5$ µm$^2$ at the apex. One section per heart at the middle of the apex was quantified and presented in *Supplementary file 1c*.

## Zebrafish cardiac injury and tissue collection

Adult zebrafish heart resection was conducted with the *Tg(−4.9sox10:eGFP)* or *Tg(sox10:mRFP)* line, according to published protocols (*Poss et al., 2002*). Resected and sham operated fish hearts (n = 24) were collected at 7 days post injury (dpi) (n = 18), and 21 dpi (n = 53) at which time the fish were euthanized and the hearts were removed for further analysis. The hearts were fixed in 4%PFA in PBS for overnight at 4°C prior to processing for staining. Adult zebrafish were maintained in the Beckman Institute Zebrafish Facility at Caltech, and all animal and embryo work were completed in compliance with California Institute of Technology Institutional Animal Care and Use Committee (IACUC) protocol 1764.

## Immunohistochemistry and image analysis

After cryosectioning, slides were incubated in 1xPBS at 42°C to remove gelatin. 0.3% vol/vol Triton-X100in 1xPBS was used to permeabilize the tissue. Sections were incubated with primary antibody underneath a parafilm layer at 4°C overnight (primary antibody dilutions: 1:10 Troponin T CT3, DSHB (RRID:AB_528495); 1:10 Myosin Heavy Chain ALD58, DSHB (RRID:AB_528361); 1:10 Myosin Heavy Chain F59, DSHB (RRID:AB_528373); 1:500 Mouse anti-smooth muscle actin, Sigma-Cat# A5228-200uG; 1:500 Mouse anti phospho-histone H3, Abcam-ab14955; 1:500 rabbit anti caspase-3,R and D

Systems # AF835; 1:500 goat anti GFP, Abcam Cat#ab6673, all in blocking reagent 1xPBS with: 5% vol/vol normal donkey serum, 0.3% vol/vol Triton-X100). Subsequently, sections were washed for 3 times with 1xPBS, incubated with secondary antibody for 40 mins at room temperature and counter-stained with DAPI. Secondary antibodies include: Goat anti-mouse IgG2a Alexa-568 (RRID:AB_2535773), Goat anti-mouse IgG1 Alexa-568(RRID:AB_2535766), Goat anti-rabbit IgG Alexa-568 (RRID:AB_2534121), Donkey anti-goat IgG Alexa-488(RRID:AB_2534102); 1:1000, Molecular Probes. Zeiss AxioImager.M2 with Apotome.2 and Zeiss LSM 800 confocal microscope were utilized for imaging. Images were cropped, rotated, and intensity was linearly adjusted for visualization.

### In situ hybridization of adult zebrafish hearts

After fixation, hearts were embedded in paraffin and sections were prepared at 10 μm thickness on a Zeiss microtome. After paraffin removal with histosol, sections were washed and then hybridized with 1 ng/μl anti-sense digoxygenin-labeled probes overnight at 70℃ in a humidifying chamber. After hybridization, sections were washed with 50% formamide/50% 1X SSCT buffer followed by washes with MABT and a blocking step in 1% Roche blocking reagent. Sections were then incubated overnight at room temperature with a 1:2000 dilution of anti-DIG-Alkaline Phosphatase antibody (Roche). After several washes with MABT, chromogenic color was developed using NBT/BCIP precipitation (Roche).

### Transcriptome analysis of regenerating neural crest cells

For each replicate (n = 2), regenerating ventricles (n = 12) were isolated at 21 days post injury and dissociated into a single cell suspension using a pestle-A tissue homogenizer followed by incubation in Accumax (Innovative Cell Technologies, Inc) at 30℃. sox10-mRFP-positive and sox10-mRFP-negative cells were collected by FAC-sorting on a BD Biosciences FACSAriaFusion Cell Sorter. cDNA from mRFP-positive and negative cells was prepared using SMART-seq Ultra Low Input RNA Kit V4 (Takara) according to the manufacturer's protocol. Sequencing libraries were built according to Illumina Standard Protocols and sequenced using an Illumina HiSeq2500 sequencer at the Millard and Muriel Jacobs Genetics and Genomics Laboratory (California Institute of Technology, Pasadena, CA). 50 million, 50 bp, single-ended reads from two biological replicates were mapped to the zebrafish genome (GRCz10) using Bowtie2 (*Langmead and Salzberg, 2012*). Transcript counts were calculated using featureCounts (Subread) and differential gene expression analysis was performed using DESeq2 (*Liao et al., 2014*; *Love et al., 2014*). Protein classification analysis was performed using PANTHER (*Mi et al., 2019*). Heatmaps of normalized counts were generated using Heatmap2. Databases have been deposited to NCBI (BioProject # PRJNA526570).

## Acknowledgements

We would like to thank Drs. Xia Han and Yang Chai at University of Southern California, Center for Craniofacial Molecular Biology for being extremely supportive and kindly providing *Wnt1-Cre; ZsGreen^{fl/fl}* cardiac tissue. Many thanks to Dr. Jeffrey Bush at University of California, San Francisco who generously sent us *Wnt1-Cre2+; R26mTmG* mouse embryos. We appreciate the help from Drs. Ann M Cavanaugh and Jau-Nian Chen at Department of Molecular, Cell and Developmental biology, University of California, Los Angeles in sharing *Tg (NC: mCherry)* transgenic fish line for *Sox10: GAL4-UAS-Cre;ubi:Switch*. We would also like to acknowledge the Caltech Millard and Muriel Jacobs Genetics and Genomics Laboratory, in particular, Igor Antoshechkin for sequencing of our RNAseq libraries. We thank Rochelle Diamond and Diana Perez of the the Caltech Flow Cytometry Cell Sorting Facility for cell sorting assistance. We thank David Mayorga and Ryan Fraser of the Beckman Institute Zebrafish Facility for help with zebrafish husbandry and Joanne Tan-Cabugo and Constanza Gonzalez for technical assistance, and Beckman Institute Biological Imaging Facility for equipment. This work is supported by NIHR01DE027568 and NIHRO1HL14058 to MEB and a Helen Hay Whitney Post-doctoral Fellowship to MLM.

## Additional information

### Competing interests

Marianne E Bronner: Senior editor, *eLife*. The other authors declare that no competing interests exist.

### Funding

| Funder | Grant reference number | Author |
|---|---|---|
| National Institutes of Health | NIHR01DE027568 | Marianne E Bronner |
| National Institutes of Health | NIHRO1HL14058 | Marianne E Bronner |

The funders had no role in study design, data collection and interpretation, or the decision to submit the work for publication.

### Author contributions

Weiyi Tang, Conceptualization, Formal analysis, Validation, Investigation, Visualization, Methodology, Writing—original draft, Writing—review and editing, Conceived the project, performed virus preparation, lineage analysis in chick and mouse, immunohistochemistry, quantification, and wrote the manuscript; Megan L Martik, Conceptualization, Formal analysis, Validation, Investigation, Visualization, Methodology, Writing—original draft, Writing—review and editing, Conceived the project, performed the heart regeneration experiments and RNAseq, and wrote the manuscript; Yuwei Li, Formal analysis, Validation, Visualization, Methodology, Writing—original draft, Writing—review and editing, Performed molecular cloning for virus preparation, provided consultation for the manuscript; Marianne E Bronner, Conceptualization, Resources, Supervision, Funding acquisition, Investigation, Writing—original draft, Project administration, Writing—review and editing, Conceived the project, assisted with lineage analysis in chick, and wrote the manuscript

### Author ORCIDs

Weiyi Tang (iD) https://orcid.org/0000-0002-1279-1001
Megan L Martik (iD) https://orcid.org/0000-0003-1186-4085
Yuwei Li (iD) http://orcid.org/0000-0001-7753-4869
Marianne E Bronner (iD) https://orcid.org/0000-0003-4274-1862

### Ethics

Animal experimentation: Adult zebrafish were maintained in the Beckman Institute Zebrafish Facility at Caltech, and all animal and embryo work were completed in compliance with California Institute of Technology Institutional Animal Care and Use Committee (IACUC) protocol 1764.

### Decision letter and Author response

Decision letter https://doi.org/10.7554/eLife.47929.017
Author response https://doi.org/10.7554/eLife.47929.018

## Additional files

### Supplementary files

• Supplementary file 1. Quantification of cardiac neural crest contribution to heart development in amniotes and *sox10:eGFP+* cells in zebrafish heart regeneration. (**a, b**) Quantification of cardiac neural crest contribution to the heart in chick and mouse.Table presents virally labeled cardiac neural crest derivatives at MNC (migratory neural crest), CN-IX (cranial nerve nine), PAA (pharyngeal arch arteries), APS (aorticopulmonary septum), MYO (myocardium of ventricle) and IVS (interventricular septum) at day 1–6 and day 10 post injection in chick. The bottom part presents number of Wnt1+ cells in E15.5 *Wnt1-Cre* mouse. Percentage in parentheses represents the proportion of the

population among all NC-derived cells in cardiovascular structure (including MYO, APS, and IVS). % Neural crest contribution to ventricle, the proportion of Wnt1+ cells (including MYO and IVS) among all cells in the ventricle is about 16.8%. *Supplementary file 1b* shows the raw data of each embryo from which data in *Supplementary file 1a* was generated. (c) Quantification of *sox10:eGFP+* cells in the apex during zebrafish heart regeneration. Average number of *Sox10-eGFP+* cells per $2 \times 10^5$ $\mu m^2$ in one section through the middle of the apex of 7dpa (n = 3) and 21 dpa (n = 3) hearts after resection. Standard deviation is presented in parentheses next to the cell number. GFP expression was negligible in sham operated hearts (n = 3) at the same time points.

DOI: https://doi.org/10.7554/eLife.47929.012

• Transparent reporting form

DOI: https://doi.org/10.7554/eLife.47929.013

## Data availability

All data is available in the main text, the supplementary materials. Databases have been deposited to NCBI (BioProject # PRJNA526570).

The following dataset was generated:

| Author(s) | Year | Dataset title | Dataset URL | Database and Identifier |
|---|---|---|---|---|
| Weiyi Tang, Megan L Martik, Yuwei Li, Marianne E Bronner | 2019 | Cardiac neural crest contributes to cardiomyocytes in amniotes and heart regeneration in zebrafish | https://www.ncbi.nlm.nih.gov/bioproject/PRJNA526570 | BioProject/SRA, PRJNA526570 |

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
