## [Decision Letter]

Thank you for submitting your article "Cardiac neural crest contributes to cardiomyocytes in amniotes and heart regeneration in zebrafish" for consideration by *eLife*. Your article has been reviewed by Didier Stainier as the Senior Editor and Reviewing Editor, and three reviewers. The reviewers have opted to remain anonymous.

The reviewers have discussed the reviews with one another and the Reviewing Editor has drafted this decision to help you prepare a revised submission.

As noted by all three reviewers, the manuscript addresses an important issue, that of neural crest contribution to the myocardium, and provides interesting and significant new findings. While some experiments were suggested that would strengthen the paper, the consensus from the discussions was that they were not essential.

However, as detailed in reviewer #2's comments, the interpretation of some of the zebrafish data needs to be adjusted.

I have pasted below the full reviews as all the comments should be taken into account when revising the manuscript.

*Reviewer #1:*

The current manuscript reports interesting findings on the role of the cardiac neural crest cells to the development of the heart in amniotes. Using a series of elegant lineage tracing analysis in chicken and mouse embryos Bronner and colleagues describe a previously unknown contribution of the cranial neural crest cells to cardiomyocytes of the outflow tract and ventricle in vertebrates. The analyses are well performed and provide interesting data. Remarkably, the authors also report in a model with heart regenerative capacities, the adult zebrafish, a reactivation of embryonic programs related to the neural crest development upon a cardiac injury (both by in situ hybridization and transcriptome analysis). Then they speculate that the cranial neural crest-derived cardiomyocytes of the ventricles could contribute to the regenerative process in the injured adult heart. While these speculations are certainly relevant, the manuscript would benefit from further analysis to support this proposal.

Essential revisions:

- Since both proliferation and trans-differentiation are prominent processes during heart regeneration, it is relevant to assess whether there is also reactivation of a proliferation gene signature (beyond gata4) and early cardiac transcriptional regulators such as Nkx2.5, *Mef2* and Tbx5 in the neural crest-derived cardiomyocytes in response to injury.

- Can the authors comment on why have they decided to do the transcriptomic analysis at 21 dpi instead of 7 dpi, which is the early onset of reactivation of the developmental neural crest program?

- In order to rule out an epicardial origin (this cell population is also reactivated upon cardiac injury and re-expresses gene developmental programs (Cahill et al., 2017)), it is also relevant to determine whether or not the neural crest-derived cardiomyocytes express bona fide epicardial markers such as Tbx18 or Tcf21.

- Given that the neonatal mammalian heart is also capable of regeneration after various injury models (apical resection and coronary ligation injuries, see Porrello et al., 2011), assessing the reactivation of neural crest gene regulatory programs in mice would further support the most exciting conclusions of this study.

*Reviewer #2:*

In this manuscript, Tang, Martik, and colleagues examine the contributions of the cardiac neural crest to the myocardium. This is an important and controversial topic with substantial relevance to the causes of congenital heart disease and to the evolution of neural crest derivatives, as well as potential relevance to mechanisms underlying tissue regeneration. It is therefore valuable that the authors provide new lineage analysis data sets that document the contribution of some cardiac neural crest cells to the myocardium in both chick and mouse. This work is carefully performed and interpreted and provides new information of interest to the field. In addition, the authors demonstrate that sox10, as well as other genes associated with the migratory neural crest, are upregulated in the myocardium of the regenerating zebrafish heart, following injury. These new gene expression data add to our understanding of how the adult zebrafish heart responds to injury, and the authors are interested in drawing a connection between these observations and the contributions of cardiac neural crest-derived cells to regeneration. These data are also valuable to those seeking to understand the regulation of cardiac regeneration, but would benefit from some interpretative adjustments. Specifically:

1) The induction of expression of sox10 and other neural crest-associated genes is very interesting, but it is inappropriate to interpret this as evidence that cardiac neural crest cells contribute to the regenerative capacity of the zebrafish heart without evidence that the cells activating expression of these genes are indeed neural crest-derived. The injury stimulus may induce expression of sox10 et al. in cardiomyocytes that were not originally neural crest-derived, and a different type of lineage tracing experiment would be required to determine whether neural crest-derived cardiomyocytes indeed respond to injury differently than lateral mesoderm-derived cardiomyocytes do. Alternatively, the authors could amend their interpretation – these changes in gene expression are certainly interesting whether or not the responding cells are necessarily neural crest-derived. (Saying that the sox10:GAL4-UAS-Cre line permanently labels all neural crest lineages is an overstatement – it permanently marks sox10-expressing cells, but there may be instances of sox10 expression outside of the neural crest.)

2) Along the same lines, the data shown here associate the induction of sox10 expression (and the expression of other neural crest-associated genes) with regeneration but do not directly demonstrate the significance of these genes to the regenerative response. Demonstrating their significance would surely require substantial additional experimentation; here, the authors should simply take caution to avoid implying that these particular genes are associated with regulation of regeneration.

3) Could the authors comment on the relationship of the differential gene expression that they document in Figure 4H with other published datasets that examine the transcriptomic changes triggered by injury? Is it the case that the same neural crest-associated genes are differentially expressed in those datasets (but haven't been highlighted in the analysis done thus far)?

*Reviewer #3:*

This is an interesting paper showing additional contributions of the neural crest to parts of the heart across vertebrates. Moreover, the authors present evidence that during regeneration of the zebrafish heart, neural crest-derived heart cells contribute to the regenerate and redeploy a neural crest gene expression programme. This suggests previously unappreciated heterogeneity in the cells contributing to regeneration.

Overall, the findings are important, but the manuscript could be substantially improved by more explanations and quantifications of data.

1) The manuscript has two parts and it is not entirely clear how these are connected. In the first part, the authors use lineage tracing to find new and known populations of crest cells that contribute to heart development. These cells seem to be rare in the apex "…"no neural crest-derived cardiomyocytes were observed in the apex of the heart"…"). However, in the second part of the manuscript they remove the apex for regeneration purposes. How are the two parts of the manuscript connected?

2) The authors say that they have found novel crest-derived cell populations in the heart. At the same time, they state that previous research was controversial, and tools were inadequate due to high background. It is unclear whether previous papers already described these populations and this is here confirmed with more robust tools, or whether these populations are entirely new. Moreover, it would be good to at least discuss in more detail how they know that their tools are superior in their hands (e.g. "the improved Wnt1 line").

3) Many results rely on scoring for the presence or absence of cells. How were tissues scored? More quantifications of the relative contribution to the tissue in question should be shown for lineage tracing and zf heart regeneration, such that the relative importance of the crest contribution can be appreciated.

4) Figure 4—figure supplement 1: Were there also cells that expressed sox10 mRNA de novo? The prediction would be that only lineage-traced cells re-express sox10 – otherwise non-neural crest derived cells contribute to the population of sox10+ cells in regeneration. Can this be clearly observed in the combination of colorimetric ISH and transgenic fluorescence?

5) More information is needed on how the "the cardiac neural crest migratory gene regulatory module" was identified in the expression profile of regenerated heart cells in zf. e.g. did they use GO analysis? They highlight 20 out of 656 genes – what are the other genes? Why is 21 days post-injury the best time point?

---

## [Author Response]

Reviewer #1:

[…] Remarkably, the authors also report in a model with heart regenerative capacities, the adult zebrafish, a reactivation of embryonic programs related to the neural crest development upon a cardiac injury (both by in situ hybridization and transcriptome analysis). Then they speculate that the cranial neural crest-derived cardiomyocytes of the ventricles could contribute to the regenerative process in the injured adult heart. While these speculations are certainly relevant, the manuscript would benefit from further analysis to support this proposal.Essential revisions:- Since both proliferation and trans-differentiation are prominent processes during heart regeneration, it is relevant to assess whether there is also reactivation of a proliferation gene signature (beyond gata4) and early cardiac transcriptional regulators such as Nkx2.5, Mef2 and Tbx5 in the neural crest-derived cardiomyocytes in response to injury.

Thank you for this suggestion. We now include information about genes associated with cell proliferation as well as early cardiac transcription factors, including Nkx2.5, Gata4 and Tbx5 in our dataset. From our RNA-seq data, we do not see enrichment of these factors compared to injured, sox10-negative cells. While not enriched, these genes are expressed at low levels, perhaps because of the time point examined.

To address the reviewer’s comment, we have included a heatmap of normalized counts in our supplemental data (Figure 4—figure supplement 2) to show enrichment of known regeneration genes in our 21dpa dataset. As suggested by the reviewer, the RNAseq data reveal expression of cell cycle and proliferation genes, consistent with an important role for proliferation of sox10+ cells. We now include data on enrichment of cell cycle genes in Figure 4—figure supplement 2.

- Can the authors comment on why have they decided to do the transcriptomic analysis at 21 dpi instead of 7 dpi, which is the early onset of reactivation of the developmental neural crest program?

We selected the 21 day time point for practical reasons, since we noted a significant number of Sox10-expressing cells in the heart at this time after regeneration as opposed to fewer cells at 7 days. We validated gene expression by in situ hybridization at both time points but focused the RNA-seq to the later time point. Moreover, at 21 days, we see the full progression of neural crest gene regulatory states from induction to migration that may have been missed at an earlier time point. However, the reviewer raises an excellent point and we will try to profile more time points for future experiments.

- In order to rule out an epicardial origin (this cell population is also reactivated upon cardiac injury and re-expresses gene developmental programs (Cahill et al., 2017)), it is also relevant to determine whether or not the neural crest-derived cardiomyocytes express bona fide epicardial markers such as Tbx18 or Tcf21.

Thank you for this suggestion. To address this possibility, we have looked through our datasets for early cardiac transcription factors. The results show epicardial markers such as Tbx18 or Tcf21 are not enriched in our Sox10+ cells compared to Sox10- cells in the regenerating heart.

- Given that the neonatal mammalian heart is also capable of regeneration after various injury models (apical resection and coronary ligation injuries, see Porrello et al., 2011), assessing the reactivation of neural crest gene regulatory programs in mice would further support the most exciting conclusions of this study.

We agree with the reviewer that it would be logical to extend this to neonatal mice. As this will require learning a new injury model, we would prefer to pursue this in a future publication, but have added this as a discussion point to be addressed in the future.

Reviewer #2:

[…] These new gene expression data add to our understanding of how the adult zebrafish heart responds to injury, and the authors are interested in drawing a connection between these observations and the contributions of cardiac neural crest-derived cells to regeneration. These data are also valuable to those seeking to understand the regulation of cardiac regeneration, but would benefit from some interpretative adjustments. Specifically:1) The induction of expression of sox10 and other neural crest-associated genes is very interesting, but it is inappropriate to interpret this as evidence that cardiac neural crest cells contribute to the regenerative capacity of the zebrafish heart without evidence that the cells activating expression of these genes are indeed neural crest-derived. The injury stimulus may induce expression of sox10 et al. in cardiomyocytes that were not originally neural crest-derived, and a different type of lineage tracing experiment would be required to determine whether neural crest-derived cardiomyocytes indeed respond to injury differently than lateral mesoderm-derived cardiomyocytes do. Alternatively, the authors could amend their interpretation – these changes in gene expression are certainly interesting whether or not the responding cells are necessarily neural crest-derived. (Saying that the sox10:GAL4-UAS-Cre line permanently labels all neural crest lineages is an overstatement – it permanently marks sox10-expressing cells, but there may be instances of sox10 expression outside of the neural crest.)

In the revised manuscript, we raise the caveat that we cannot rule out the possibility that these cells are reactivating Sox10 and the neural crest program but may come from another lineage.

In support of a neural crest origin, Abdul-Wajid, 2018 have also shown that ablation of the embryonic neural crest yields few or no sox10+ cells in the adult heart. This suggests there are no post-embryonic neural crest additions to the heart. This is now discussed.

On a personal note, I find it very unlikely that the Sox10+ cells would come from another lineage. To my knowledge, there are no non-neural crest-derived Sox10-expressing cells in the periphery. There are a few Sox10+ non-neural crest-derived cells in the CNS, but they are confined to the brain. Therefore, I strongly suspect that these are indeed neural crest-derived. That said, I agree that this would need to be more rigorously validated and therefore we are more circumspect in the conclusions drawn in the manuscript.

2) Along the same lines, the data shown here associate the induction of sox10 expression (and the expression of other neural crest-associated genes) with regeneration but do not directly demonstrate the significance of these genes to the regenerative response. Demonstrating their significance would surely require substantial additional experimentation; here, the authors should simply take caution to avoid implying that these particular genes are associated with regulation of regeneration.

We agree that a much more detailed analysis would be necessary to directly demonstrate mechanism. We have amended the manuscript to make this point and be more circumspect.

3) Could the authors comment on the relationship of the differential gene expression that they document in Figure 4H with other published datasets that examine the transcriptomic changes triggered by injury? Is it the case that the same neural crest-associated genes are differentially expressed in those datasets (but haven't been highlighted in the analysis done thus far)?

By investigating the expression of our *sox10*+ enriched genes on RegenDbase.org (an online database of both published and unpublished regeneration transcriptome analyses), we find many of our enriched genes are present at 21dpa, but not statistically significant when compared to the 0dpa vs 21dpa dataset.

Reviewer #3:

This is an interesting paper showing additional contributions of the neural crest to parts of the heart across vertebrates. Moreover, the authors present evidence that during regeneration of the zebrafish heart, neural crest-derived heart cells contribute to the regenerate and redeploy a neural crest gene expression programme. This suggests previously unappreciated heterogeneity in the cells contributing to regeneration.Overall, the findings are important, but the manuscript could be substantially improved by more explanations and quantifications of data.1) The manuscript has two parts and it is not entirely clear how these are connected. In the first part, the authors use lineage tracing to find new and known populations of crest cells that contribute to heart development. These cells seem to be rare in the apex "…"no neural crest-derived cardiomyocytes were observed in the apex of the heart"…"). However, in the second part of the manuscript they remove the apex for regeneration purposes. How are the two parts of the manuscript connected?

Thank you for pointing out this issue. We have tried to integrate the two parts of the manuscript better. Basically, given that neural crest cells are a well-known stem cell population, the finding that the they can contribute to cardiomyocytes in amniotes similar to what had been previously shown in anamniotes made us wonder if they might contribute to heart regeneration. Zebrafish seemed like the most appropriate model for this due to their extensive capacity for heart regeneration.

To improve the connection between the two parts of the paper, we now more thoroughly reference and discuss several previously published papers that use the sox10 promoter for labeling cardiac neural crest cells (Carney et al., 2006; Cavanaugh et al., 2015; Abdul-Wajid et al., 2018). These studies show that in zebrafish, the neural crest contribution to the ventricles is primarily to the trabecular myocardium, similar to what we have found in chick and mouse. The study by Cavanaugh and colleagues also shows that *sox10* is downregulated in the heart as development proceeds (Cavanaugh et al., 2015). We confirm this by showing minimal *sox10* expression in control adult hearts.

The finding that *sox10-*derived cells are primarily in the proximal trabecular myocardium of the zebrafish heart suggests that these cells must be migrating into to the wound site after injury. We now discuss this interesting possibility more clearly.

2) The authors say that they have found novel crest-derived cell populations in the heart. At the same time, they state that previous research was controversial, and tools were inadequate due to high background. It is unclear whether previous papers already described these populations and this is here confirmed with more robust tools, or whether these populations are entirely new. Moreover, it would be good to at least discuss in more detail how they know that their tools are superior in their hands (e.g. "the improved Wnt1 line").

Thank you for pointing out these issues. We have added further discussion of why the contribution to cardiomyocytes may have been missed in birds and previous hints from the literature of a neural crest contribution to the mammalian heart. In birds, this population was totally missed. There is no mention of a contribution of neural crest cells to cardiomyocytes in any of the quail/chick chimera papers. In mice, the lineage tracing was done with P0 mice which is not a neural crest specific marker, but rather labels Schwann cells and notochord cells, making these results inconclusive at the time. In fact, Fukada and colleagues recently published a paper (Fukada et al., 2017) describing the differences between P0 and Wnt1 lineage labeling in mice and described Wnt1 as the “gold standard” for neural crest labeling.

In our study, we use two different Wnt1 lines to examine the contribution of neural crest cells to the murine heart. The reason for using the “improved” Wnt1 line (*Wnt1-Cre2+: R26mTmG)* is because the original line (*Wnt1-Cre:ZsGreen)*, used in most publications, is known to have ectopic expression in some locations due to *Wnt1* activation caused by an in-frame ATG located upstream of *Wnt1* start codon, thus complicating the results. The Wnt1-Cre2+ corrects this issue and is therefore a more reliable reporter. We think that previous authors missed the neural crest contribution due to the myocardium in Wnt1 mice because the early experiments done with Wnt1-cre lineage tracing used colorometric lacZ as a readout. As there is high background from endogenous enzyme activity in the heart, it is likely that any labeling observed in this location was ignored as high background. Thus, with our tools, including retroviral lineage tracing in chick and fluorescent Wnt1-cre2+ mouse lines, we both confirmed all other previously described contributions of cardiac neural crest to the heart (Kirby et al., 1983, Jiang et al., 2000) as well as discovering a novel contribution to the myocardium. We have clarified this point in the paper and thank the reviewer for point out this confusion.

3) Many results rely on scoring for the presence or absence of cells. How were tissues scored? More quantifications of the relative contribution to the tissue in question should be shown for lineage tracing and zf heart regeneration, such that the relative importance of the crest contribution can be appreciated.

We thank the reviewer for the suggestion. To address this, we quantitated the numbers of labeled neural crest cells we observe in various cardiac locations. As we do not label the entire neural crest population in chick, it’s most accurate to compare the% of labeled cells in each location (explained Supplementary file 1 legend). For the mouse, we label the entire neural crest population and we find that 16.8% of the labeled neural crest cells are in the proximal ventricles (quantified by automated particle analysis in FIJI program). Supplementary file 1A presents the quantified virally labelled cells in the chick embryo and E15.5 mouse and include the average percentage of neural crest contribution in the ventricular myocardium in mouse. We clarify that tissue were scored for numbers of GFP+ cells and clarify how counting was done in sections through chick, mouse and fish tissue in the methods. We have added approximate numbers of GFP+ cells to Supplementary file 1C in the zebrafish regeneration experiments as well.

4) Figure 4—figure supplement 1: Were there also cells that expressed sox10 mRNA de novo? The prediction would be that only lineage-traced cells re-express sox10 – otherwise non-neural crest derived cells contribute to the population of sox10+ cells in regeneration. Can this be clearly observed in the combination of colorimetric ISH and transgenic fluorescence?

Thanks for raising this. We now clarify that only cells that lineage trace with the sox10 marker reexpress *sox10* during regeneration. However, not all mCherry+ cells express *sox10*, presumably because they lose expression as they differentiate into cardiomyocytes. We now include an updated figure that is clearer.

5) More information is needed on how the "the cardiac neural crest migratory gene regulatory module" was identified in the expression profile of regenerated heart cells in zf. e.g. did they use GO analysis? They highlight 20 out of 656 genes – what are the other genes? Why is 21 days post-injury the best time point?

Thank you for raising this issue as we now realize that we were unclear. For comparison, we use data from a recently published paper in which we profiled migratory cardiac neural crest genes in chick (Tani-Matsuhana et al., 2018). We then looked for similar gene signatures in our zebrafish *sox10+* dataset, focusing on transcription factors associated with modules of the neural crest gene regulatory network (Martik and Bronner, 2017). We have also highlighted known zebrafish neural crest genes on our volcano plot (Figure 4D). This gene list was acquired by a literature search as well as ZFIN. We now clarify this in the text.